# Methodology for developing and validating Bangladesh healthy eating index: A study protocol

Ahmed Jubayer[1,2], Abira Nowar[1], Saiful Islam[1]\*, Md. Hafizul Islam[1], Md. Moniruzzaman Nayan[1]

1 Institute of Nutrition and Food Science, University of Dhaka, Dhaka, Bangladesh, 2 Bangladesh Institute of Social Research (BISR) Trust, Dhaka, Bangladesh

\* saifulinfs@du.ac.bd

## Abstract

### Background

One of the widely used methods for evaluating the overall quality of a diet is the Healthy Eating Index. In the present study protocol, we lay out the methodological approach to the development and validation of a healthy eating index specific for the Bangladeshi population (hereinafter referred to as Bangladesh Healthy Eating Index (BD-HEI)).

### Methods

Bangladesh Healthy Eating Index will be developed based on the Food-based dietary guidelines (FBDG) of Bangladesh. Using a scoring system that aligns with the eleven food groups outlined in the FBDG, the index will consist of eleven eating components. A cross-sectional community nutrition survey will be carried out among 1080 reproductive-aged women. Through this survey, quantitative dietary data will be collected following multiple pass 24h dietary recall method. Repeated 24-hour dietary data (of two consecutive days) will be collected from one-third of the respondents. Evaluating usual food and nutrient intake as well as the probability of nutrient intake adequacy, the study will examine the validity of the BD-HEI. Following a suitable statistical procedure, the reliability and construct validity of BD-HEI will be evaluated.

### Significance of the study

Country-specific HEI can be used to assess the dietary quality of the people of that country. The findings from this research can inform policy decisions and strategies to promote healthier eating habits and combat the rising burden of diet-related diseases in the country.

**Data Availability Statement:** Deidentified research data will be made publicly available when the study is completed and published.

**Funding:** The author(s) received no specific funding for this work.

**Competing interests:** The authors have declared that no competing interests exist.

## Background

Dietary pattern analysis is a promising approach for understanding diet-health and diet-disease complex relationships. It has been suggested that focusing on dietary patterns rather than individual foods or nutrient intake can help to better understand the relationship between diet and non-communicable diseases (NCDs) [1].

Food-based dietary guidelines (FBDGs) are developed as specific, culturally appropriate, and actionable recommendations based on scientific evidence on the relationships between diets and health outcomes. These guidelines, influence consumers' dietary behavior, can serve as the basis for a number of national food, nutrition, and health-related policies and programs [2]. With the aim to achieve various national goals and ensure healthy diets for all, the government of Bangladesh updated its FBDG in 2020. This latest FBDG provides more quantitative information than before and specifies serving size recommendations. To evaluate the potential impact of such FBDGs, it is important to study the relationships between adherence to the FBDGs recommendations and the associated health outcomes [3].

One approach to assessing adherence to the FBDGs is to construct a priori dietary quality index based on national or international dietary recommendations. The Healthy Eating Index (HEI) is one such dietary quality index that measures how closely a certain meal combination follows established nutritional recommendations [4, 5]. HEI has several components of adherence to dietary guidelines for each food group or specific foods or nutrients and the ratings are assigned to each component. A total score for HEI is calculated by adding the scores for each component, which demonstrate compliance with dietary recommendations.

HEI can be used for assessing diet quality at the population level and monitoring it over time [6]. Furthermore, nutritional epidemiologists can investigate the diet-disease associations and the risk of mortality through the application of HEI [7]. Many countries including Japan [8], Netherlands [1], Ethiopia [4], and Vietnam [5] have developed a Healthy eating index (HEI) based on their current FBDGs.

The latest FBDG for Bangladesh provides guidelines on balanced diet, limiting salt and sugar, moderate use of fat, and special care for pregnant and lactating mothers [9]. Country-specific HEI is required to compare diet quality across subpopulations, examine dietary patterns, and explore diet-disease linkages that can guide public health interventions and policy action. However, unlike many other countries, there is currently no index in use that analyzes adherence to these FBDGs to evaluate the diet quality of the Bangladeshi population. This protocol aims to describe the methodological approach to the development and validation of the Bangladesh healthy eating index (BD-HEI) as an indicator of dietary quality in terms of adherence to the latest FBDG.

## Methods

### Components of BD-HEI

BD-HEI will be developed based on the updated FBDG of Bangladesh (Fig 1), prepared by the Ministry of Food and the Ministry of Health and Family Welfare of the Government of Bangladesh [10]. BD-HEI will comprise eleven eating components with a scoring system that corresponds to the eleven food groups listed in the FBDG. Each component of the BD-HEI will be categorized into adequacy, optimum, or moderation components based on the healthier options provided in the FBDG. For example, fruits and vegetables can be categorized as adequate components since higher intakes of these food items reduce the risk of micronutrient deficiencies. While, fish/meat/milk can be categorized as optimum components since they have both positive and negative health effects, depending on the amount consumed. Before

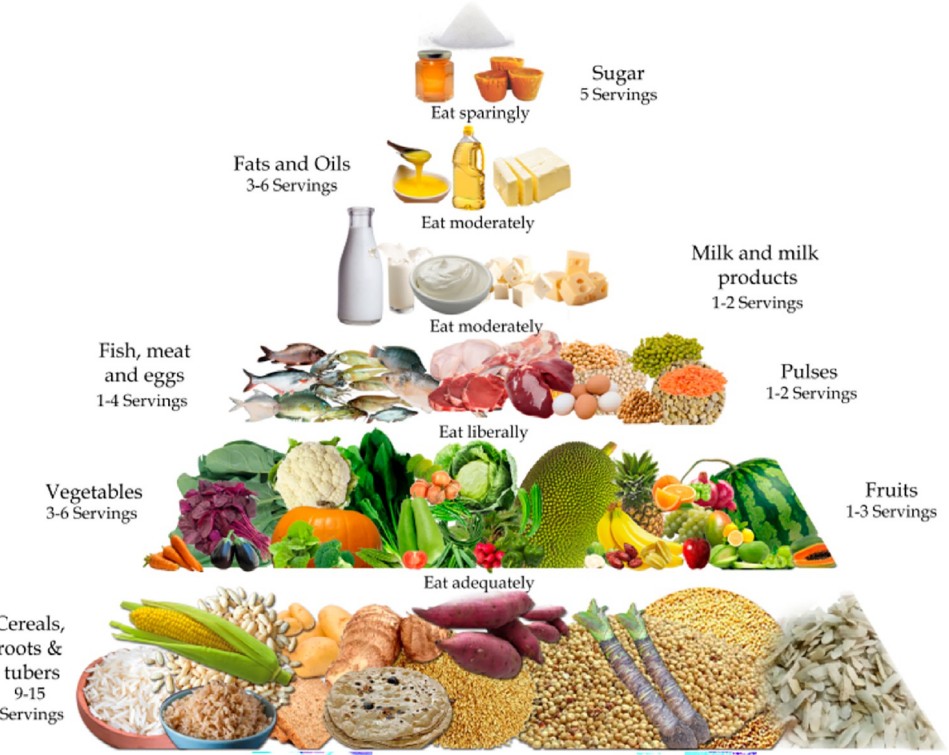

**Fig 1. Food guide pyramid for Bangladeshi population (Prepared by FPMU, GoB).**

deciding on a scoring system, foods will be assigned to food groups as they are in the FBDG. Table 1 provides a summary of the eleven components of the BD-HEI.

## Weighing, standard, and scoring

Each component will be equally weighted with a minimum score of zero and a maximum score of 5 or 10. Since all the components are equally important in the dietary guideline, each

**Table 1. Components of BD-HEI and their minimum and maximum score.**

| Dietary component | Type of scoring | Maximum score | Dietary Guidelines of Bangladesh | | Standard for minimum score (= 0) | Standard for maximum score |
|---|---|---|---|---|---|---|
| | | | Serving | Amount in gm | | |
| Cereal | Optimum | 5 | 9–14 serving/d | 270-420gm | 0 or > threshold value | 270-420gm |
| Roots, and tubers | Optimum | 5 | 1 serving/d | 30 gm | 0 or > threshold value | 30 gm |
| Pulse | Optimum | 10 | 1–2 servings/d | 30–60 gm | 0 or > threshold value | 30–60 gm |
| Fish/meat/ egg | Optimum | 10 | 1–4 servings/d | 100–400 gm | 0 or > threshold | 100–400 gm |
| Milk and milk products | Optimum | 10 | 1–2 servings/d | 150-300gm | 0 or > threshold | 150-300gm |
| Fats and oil | Optimum | 10 | 3–6 servings/d | 15–30 gm | 0 or > threshold | 15–30 gm |
| Fruits | Adequate | 10 | 1–3 servings/d | 100–300 gm | 0 | ≥100 gm/day |
| Leafy Vegetables | Adequate | 5 | 1–2 servings/day | 150–300 gm | 0 | ≥150 gm/day |
| Non- leafy Vegetables | Adequate | 5 | 2–4 servings/day | 300–600 gm | 0 | ≥300 gm/day |
| Sugar | Moderation | 10 | Less than 5 servings/ day | 25 gm | >threshold | ≤25gm/day |
| Salt | Moderation | 10 | Less than 1 servings/ day | 5gm | >threshold | ≤5g/day |
| Total | | 90 | | | | |

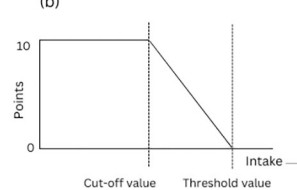
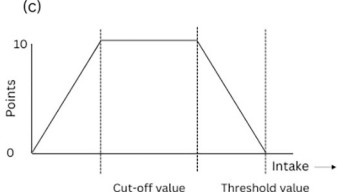

**Fig 2.** Graphical presentation of BD-HEI for (a) adequacy category, (b) moderation category, and (c) optimum category. This figure is adapted from Looman et al. 2017 [12].

component will receive equal weightage. Within the framework of the Food-Based Dietary Guidelines (FBDG), the vegetable category is subdivided into two distinct sub-components. Consequently, the overall score of 10 will evenly be allocated between these two components. Similarly, cereals, roots, and tubers are categorized together in the food guide pyramid, although there are separate recommendations for cereals and roots/tubers. Thus, the total score (10) will be allocated proportionally. A score of 10 will be assigned to each of the remaining nine components. Similar approaches (weightage and scoring distribution within food groups) were followed while updating US HEI-2015 [11]. Therefore, the total BD-HEI score will be between 0 and 90, with higher scores indicating better diet quality. There will be three distinct rating systems for three different categories (i.e., adequacy, moderation, and optimum) of components. Fig 2 and Table 1 illustrate the scoring system for three distinct HEI components.

**Scoring for the adequacy component.** Food groups such as fruits and vegetables should be consumed adequately. These food groups are considered healthy, and higher score will be allotted for higher consumption of foods from these groups. Participants will receive a minimum score of zero if they did not consume any item from this category. A maximum score of 5 or 10 will be given if they consume the recommended serving size or greater. For the fruits group, the maximum score will be 10. While, for the leafy and non-leafy vegetable groups, the maximum score will be 5 for each sub-component. The following formula (slightly adapted from Bekele et al. [13]) will be used to compute the score when participants eat fewer portions than advised:

$$\text{Maximum score (5 or 10)} \times \frac{\textit{Consumed amount of serving}}{\textit{Recommended amount of serving}}$$

**Scoring for the optimum component.** Food groups such as cereals and fish/meat/eggs should be consumed within an optimal range. A maximum value (five or ten) will be allotted if intake is within the recommended range. Zero will be set as the minimum score when either there is no consumption of any item from a food group or consumption exceeds the highest consumption limit (threshold value). If the threshold value is not found in FBDG, it will be set as the 85th percentile of the average intake of the sample population [13, 14].

When consumption is less than the lower limit of the recommended range, the score will be computed with the following formula:

$$\text{Maximum score (5 or 10)} \times \frac{\textit{Consumed amount of serving}}{\textit{Lower limit of the recommended range}}$$

If consumption exceeds the upper limit of the recommended range, the following formula will be followed:

$$\text{Maximum score } (5 \text{ or } 10)$$
$$- \frac{(\textit{Consumed amount of serving} - \textit{the upper limit of the recommended range}) \times \max(5 \textit{ or } 10)}{\textit{Upper limit of the recommended range}}$$

**Scoring for the moderation component.** Unhealthy food items will be placed into this category; thus, less consumption will carry a higher score. A maximum score (ten) will be assigned if consumption is less than the recommended servings. When participants consume more than the recommended servings, scores will be calculated using the following equation:

$$\text{Maximum score } (10)$$
$$- \frac{(\textit{Consumed amount of serving} - \textit{recommended amount of servings}) \times 10}{\textit{Recommended amount of servings}}$$

Zero will be taken when consumption exceeds the highest consumption limit (threshold value). If the threshold value is not found in FBDG, it will be set as the 85[th] percentile of the average intake of the sample population [13, 14].

## Conduction of a community nutrition survey

**Study design, sample size, and sampling.** A cross-sectional survey will be carried out in 1080 households from different rural and urban areas of Bangladesh. A two-step cluster sampling will be applied. Based on a previous study [15], the entire Bangladesh (64 districts) will be divided into four groups based on productivity index and composite score. These groups are low-productive, moderately low-productive, moderately high-productive, and highly productive regions [15]. The reasons for the choice of such classification are that dietary pattern and food security are correlated with cropping pattern, agricultural productivity [16].

From each region, one district will be selected randomly. Then, one urban and two rural areas will be randomly selected from each district. Following the Bangladesh Demographic and Health Survey (BDHS), proportionately more areas (one areas from urban and two areas from rural) will be taken from rural areas compared to urban areas. Mega city, City corporation, Paurashava/Municipality Area, and Upazila headquarters will be considered as urban area (BBS, 2014). However, each selected area will be divided into 3 Enumeration areas (EA). Therefore, in each district, there will be 9 EAs (6 from rural areas and 3 from urban areas). Thus, a total of 36 EAs will be obtained from four districts.

In the second step, a complete list of households will be prepared for each of the 36 EAs and a distinct ID number will be assigned for each household. A total of 30 households will be randomly selected from each of the 36 EAs. Thus, the survey will be conducted on 1080 households from different rural and urban areas of Bangladesh. From each household, one reproductive-aged woman will be interviewed who is primarily responsible for household food preparation. The steps involved in selecting a representative sample are depicted in Fig 3.

**Questionnaire preparation.** A well-structured survey questionnaire will be prepared for the community nutrition survey. The questionnaire has two parts: the demographic information, socioeconomic status of the household, chronic disease condition, and other relevant data will be collected through part 1 of the questionnaire. Furthermore, dietary data will be collected using a different question item (part 2). To ensure the questionnaire is suitable, a pilot survey will be carried out prior to the final data collection.

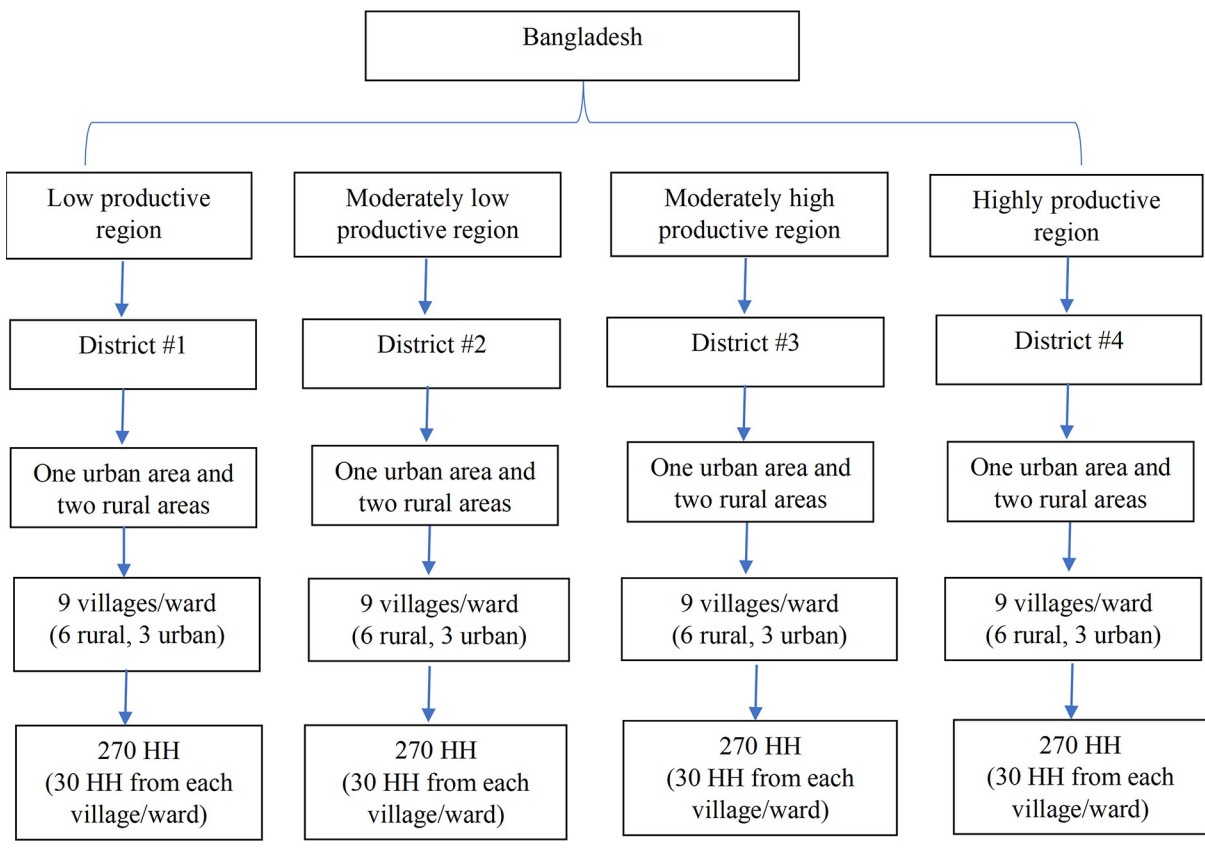

**Fig 3. Sample selection process for the cross-sectional survey.**

**Enumerator training.** The required number of supervisors and enumerators will be appointed to administer the survey. All the enumerators and supervisors will be nutrition graduates. We will arrange a comprehensive three-day training program that will consist of an in-house orientation and repeated practice sessions.

**Data collection plan.** Data will be collected utilizing an online data collection platform such as the Kobo toolbox (https://www.kobotoolbox.org/). Dietary data will be collected using a quantitative multiple-pass 24-hour dietary recall method. 24h dietary data of two consecutive days will be collected from one-third of the respondents. A woman of reproductive age who is primarily responsible for household food preparation will be asked about the cooked or raw weight of each food item or dish consumed either within or outside the home. The enumerators will carry household measuring cups of different sizes and kitchen scales to enable the respondents to recall the portion size/weight of food items they consumed during the previous 24h.

**Data analysis plan.** Data stored on the Kobo platform will be transformed into SPSS and SAS software for further analysis. A stratified two-step cluster sampling approach will be used for the sampling design, with district productivity serving as the basis for stratification and district, area, and EA level clustering. It is anticipated that each stratum and cluster will have equal odds of selection in this sampling design. As a result, straightforward random sampling inside each stratum or cluster is presumed, and intricate weighting modifications are not required. Each sampled household will receive sampling weights since a two-step cluster sampling strategy will be used. Considering the stratification and clustering in the design, the

sampling weight for every household will be computed as the inverse of its likelihood of selection. The survey results can then be appropriately represented to the target population by using these sampling weights in the analysis. Based on food intake data, usual energy and nutrient intake will be calculated using food composition table for Bangladesh [17]. For calculating usual intake distribution, Markov Chain Monte Carlo (MCMC) approach [18] will be applied and the adequacy of nutrient intake will be evaluated following the probability approach [19]. Both descriptive and inferential statistics will be computed including Principal component analysis and reliability analysis.

**Analytical approach for measuring validity and reliability of the index.**   Table 2 summarizes the strategies to be used to evaluate the construct validity and reliability of BD-HEI.

**Reliability test.**   The reliability test measures the index's internal consistency, or how the constituent components relate to one another. Two types of correlation analyses will be performed: pairwise correlation among HEI components to examine the relationship among components, the correlation between each component, and the total score of the remaining component which is referred to as item-rest correlation [20].

In addition to correlation analyses, Cronbach's alpha will be calculated to examine the internal consistency of items. Internal consistency measures the "degree of the interrelatedness among the items" [21]. It gauges the extent to which different index components measure the same underlying construct. The coefficient is impacted by the construct's dimensionality. A high value is not expected in this case because Cronbach's alpha is used under the assumption of one-dimensionality, yet the HEI is scoring a multidimensional construct.

**Construct validity test.**   Construct validity test aims to determine to what extent does the index accurately measures a healthy diet. In this case, both convergent and divergent validity will be evaluated. Convergent validity describes how closely two measures of constructs that, in theory, should be connected are related. Divergent validity determines whether two tests that ought not to be highly correlated are indeed unconnected. The results of two tests evaluating two independent constructs should have little to no correlation with one another [22]. Construct validity will be tested by four types of analyses:

*Menu scoring*: It is expected that the index assigns high scores for exemplary menus. Menu plan prepared by BIRDEM (2013) for moderately active women for a single day will be used and BD-HEI will be applied following a simple scoring method to assess whether it produces high scores.

*Index's capability of ensuring sufficient variation among individuals*: Percentiles of total score and healthy eating index component score will be computed based on dietary intake data. The total HEI score and individual component score should show sufficient variation among individuals. To account for anticipated variations in dietary quality among groups, the HEI scores should show substantial variation, lending credence to the construct validity.

*Variation of HEI across various socio-demographic group*: Analyses will focus on whether the participants' socio-demographic characteristics differ significantly throughout the tertiles of the Bangladesh Healthy Eating Index (HEI), including changes in district distribution, education levels, household food security status, dietary diversity, and wealth index. We will also look at the mean HEI score (both total score and individual component score) among different socio-demographic group. We also expect higher HEI score among women with higher dietary diversity, household food security, higher level of education, and higher wealth index.

*Measuring diet quality is independent of diet quantity*: Correlation analyses will be performed among index component score, total score, and energy intake. Weak correlation is expected. These weak correlations suggest that the BD-HEI can evaluate dietary quality apart from quantity of food consumed. This is crucial since HEI measure the diet quality which is independent of quantity of food consumed. This is an essential feature of the HEI because if

**Table 2. Strategies to be used to evaluate the construct validity and reliability of BD-HEI.**

| Item | Question | Rationale | Strategy | Expectation |
|------|----------|-----------|----------|-------------|
| **Construct validity** | | | | |
| Index's capability of ensuring sufficient variation among individuals | How much individual variation is there in the BD-HEI? | Distinguishing between individuals with varying degrees of devotion to FBDG requires scores to vary significantly. | Percentiles of total score and healthy eating index components will be computed based on dietary intake data | Most components and the total score will display the full range of possible scores across the distribution. Such findings will indicate that the BD Healthy Eating Index shows a lot of individual diversity, with a lot of different values suggesting different amounts of healthy eating adherence and population-level variations in total and component scores |
| Variation of HEI across various socio-demographic group | Does the BD-HEI distinguish between populations whose dietary quality is known to differ? | It is anticipated that an index representing FBDG adherence will yield scores that vary amongst groups with established variations in diet quality. | Compare mean HEI score across various group | We can expect higher HEI score among women with higher dietary diversity, household food security, higher level of education, and higher wealth index. Reason for such expectation that previous studies reported better diet quality among these socio-economic groups. |
| Measuring diet quality is independent of diet quantity | Is the quantity of food consumed not a factor in how the index ranks diet quality? | So that it reflects the quality and not the quantity of foods consumed, an index representing adherence to FBDG should be essentially independent of energy consumption. | Correlation analyses will be performed among index components and energy intake. | Weak correlation is expected between BD-HEI score and energy intake These weak associations suggest that the BD-HEI can evaluate dietary quality apart from quantity. This is an essential feature of the HEI because if the score was dependent on quantity of foods eaten, higher scores may be due to eating greater quantities of food rather than higher quality of foods eaten. |
| Menu scoring | Does the index yield high scores for exemplary menus? | Exemplary menus should receive high rankings from the index. | Menu plan prepared by BIRDEM (2013) for moderately active women for a single day will be used and BD-HEI will be applied following simple scoring method to assess whether it produce high scores. | A high index score indicates a high-quality diet, whereas a low score indicates a poor quality diet. |
| Multidimensionality of the index | Is the index multidimensional? | Since there are many facets to dietary choices that an index measuring adherence to FBDG evaluates, it should not rely on just one. | Principal component analysis (PCA) will be performed in dietary data to explore whether more than one factor (or item) underlies the total score. We will look at the scree plot of eigenvalues from PCA and assess how much variance is explained. | The purpose of the HEI is to assess a group of food items based on their alignment with the dietary guidelines, which offer recommendations for the whole diet. These values can be analyzed together to identify a pattern of diet quality. Additionally, a total score is calculated to represent the overall quality of the diet. No single factor should explain the total score. |
| **Reliability** | | | | |

(*Continued*)

**Table 2.** (Continued)

| Item | Question | Rationale | Strategy | Expectation |
|------|----------|-----------|----------|-------------|
| Internal consistency | How internally consistent is the score? | For the index as a whole to be useful, its component should be highly congruent with one another. | Cronbach's coefficient alpha will be calculated to measure the internal consistency | The coefficient is impacted by the construct's dimensionality. A high value is not expected in this case because Cronbach's alpha is used under the assumption of one-dimensionality, yet the HEI is scoring a multidimensional construct. The multidimensional nature of the score suggests that its internal consistency is likely to be moderate. |
| | How does the various component of the index relate to the total score? Which factors have the most impact on the total score? | | Pairwise correlation among HEI components to examine the relationship among components, correlation between each component and the total score of the remaining component which is referred to as item-rest correlation. | Pairwise correlation among index component will allow to identify the similar/ related components. It is indicative of distinct dietary components; moderate to high correlations between related components; low correlations between unrelated components. |

the score was dependent on quantity of foods eaten, higher scores may be due to eating greater quantities of food rather than higher quality of foods eaten.

In addition to energy, correlation coefficient will be calculated for several macro and micro-nutrients. In that case, both energy unadjusted nutrient intake and energy adjusted (Energy percentage usual macronutrient intake (E%), Macronutrient intake/1000kcal) will be considered.

*Multidimensionality of the index*: The purpose of the HEI is to assess a group of food items based on their alignment with the dietary guidelines, which offer recommendations for the whole diet. The HEI evaluates any combination of meals and generates separate component scores. These values can be analyzed together to identify a pattern of diet quality. Hence, it is expected that that the total score would not be explained by a single factor.

Principal component analysis (PCA) will be performed in dietary data to explore whether more than one factor (or item) underlies the total score. We will look at the scree plot of eigen-values from PCA and assess how much variance is explained.

## Ethics approval

The ethical review committee of the Faculty of Biological Sciences, University of Dhaka, reviewed and approved the protocol with reference No. 227 Biol. Scs. On August 30, 2023. Written consent will be taken from respondent prior to the interview.

## Status and timeline of the study

Draft Quantitative survey questionnaire has already been prepared. Questionnaire will be finalized after the pilot survey. We anticipate hiring the required number of enumerators by the end of October 2024, and the first week of November 2024 is when we expect to arrange the enumerator training session. The data collection process would be completed by mid-December 2024. Finally, it is anticipated that the study will be completed in the middle of 2025.

## Discussion

An excellent method for evaluating dietary quality is the healthy eating index; it measures how well a group of foods adheres to the dietary guidelines. The methodological approaches for creating and evaluating a healthy eating index for the Bangladeshi population are described in this protocol paper. Various statistical analyses will be performed to examine the psychometric properties of the index.

In recent years, Bangladesh achieved commendable progress in achieving food security. A recent study reported prevalence of food insecurity as 18.92% in rural Bangladesh [23]. Merely 33% of young children have minimum acceptable diets, while around half of the population still consume inadequate micronutrients [24]. At the same time, the double burden of malnutrition is becoming a reality. Bangladesh has also committed to the global Agenda 2030 for sustainable development, which calls for attaining SDG-1 (ending poverty), SDG-2 (improving nutrition) and SDG-3 (ensuring good health and well-being) by 2030 [25]. The nation must overcome formidable obstacles to secure food and nutrition security.

Healthy eating index would help to achieve food and nutrition security by analyzing and monitoring diet quality. It aligns with SDG-2 and SDG-3. Country-specific HEI can be used to assess the dietary quality of the people of that country. It would contribute to establishing the link between diet and non-communicable diseases (NCDs), which is crucial for population health and well-being. However, unlike many other countries, Bangladesh has no such index of its own. It is anticipated that the findings from this study can inform policy decisions and strategies to promote healthier eating habits and combat the rising burden of diet-related diseases in the country. It can help policymakers and public health officials understand the current state of dietary habits and identify areas in need of improvement.

With the view to achieving SDG-3, the government of Bangladesh has undertaken and implemented several interventions prioritizing healthy diet and proper nutrition. The proposed BD-HEI would serve as a valuable indicator of assessing diet quality of Bangladeshi population in terms of their adherence to the FBDG. BD-HEI can be used as a tool to examine the relationship between diet quality and health related outcomes along with affordability of healthy diets. Moreover, the index can also be used for creating awareness among the general population regarding adherence to FBDG and designing national nutritional programs focusing on improving dietary quality. The methodological approaches for creating a healthy eating index based on FBDG were outlined in the study protocol. To improve the validity and reliability of the index, it might be necessary to reclassify the dietary groups while assessing the metrics. For instance, even though cereal and roots/tubers are considered two sub-components in this protocol, it could be required to combine them. For veggies that are not green or leafy, similar procedures could be needed. If these actions are done, the country's food guide pyramid will still be followed.

We have also outlined some expected outcomes for each evaluation metrics in Table 2 based on previously validated HEI for other countries. The BD-HEI will also be considered a valid and reliable index if these expectations came true while the evaluative study was being conducted. For example, an energy-independent index is necessary because HEI measures quality aspects of diet rather than quantity aspects. Finally, the BD-HEI must meet certain fundamental requirements of the previously established validity and reliability procedure.

The methodological approaches described in this protocol for calculating each sub-component score are adapted from Looman et al.; 2017 [12]. However, it's may not always be the case, though, that these strategies will function flawlessly. If the planned calculation approach yields subpar results, it can be necessary to alter the calculation process while evaluating the index. An alternate method of calculation would be as follows: the highest possible score for someone who followed the suggested amount of servings from each food group would be 5 or 10. On

the other hand, a person who does not eat the recommended amounts of a food group will at least obtain a score of zero. Participants would receive scores corresponding to their intake, which would fall between the maximum and lowest amounts. For instance, a person would receive a score of 4.17 if they consumed five serves of a particular food category, but they would have needed to consume six for a maximum score. This approach was followed in Australian HEI [26].

We are expecting a weak association between HEI score and energy intake. However, many components are scored in the absolute scale (such as fruits, sugar) and scoring on the absolute scale might show a strong association with energy intake, because the amount of food eaten influences the probability of eating more or less than (absolute) thresholds. By using a density method, the relationship between energy intake and diet quality indices is weakened [11, 27], resulting in an indicator that is less dependent on the quantity of food consumed. A higher sensitivity to detect changes over time because of fewer "ceiling" and "floor" effects and uniformity of scoring standards among index components—which can be computed as ratios of all foods or beverages consumed or as a percentage of total energy (%E)—are further benefits of employing the density approach. Therefore, these dietary components could also be evaluated using relative methods in addition to absolute ones, such as percentage of total energy or food group intake per 1000 kcal.

## Limitations of the study design

The study will not cover the whole country. Besides, we will validate the index only for reproductive- aged women. Therefore, further study is needed to assess the applicability of the index for different subgroups across various regions of the country.

## Dissemination plan

Dissemination plans include peer-reviewed publications, policy briefs, workshops, social media platforms, and newspaper publications. We have a plan to arrange a workshop with targeted audiences such as academicians, policymakers, practitioners, and university students.

## Conclusion

The healthy eating index is an invaluable way to evaluate diet quality. Realizing the importance of country- specific heathy eating index for measuring overall diet quality and diet-disease relationship, the present study aims to develop a healthy eating index specific for Bangladesh (BD-HEI) based on food -based dietary guidelines. A nationally representative cross-sectional survey is planned for this purpose and following suitable statistical procedure, reliability and construct validity of this index will be measured.

## Author Contributions

**Conceptualization:** Ahmed Jubayer, Saiful Islam, Md. Hafizul Islam.

**Methodology:** Ahmed Jubayer, Saiful Islam.

**Project administration:** Abira Nowar, Md. Moniruzzaman Nayan.

**Resources:** Md. Moniruzzaman Nayan.

**Visualization:** Abira Nowar.

**Writing – original draft:** Ahmed Jubayer, Md. Hafizul Islam.

**Writing – review & editing:** Ahmed Jubayer, Abira Nowar, Saiful Islam, Md. Hafizul Islam, Md. Moniruzzaman Nayan.

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
