## [Decision Letter · Decision Letter 0]

12 Jul 2024

PONE-D-24-15594Methodology for developing and validating Bangladesh Healthy Eating Index: A Study protocolPLOS ONE

Dear Dr. Islam,

Thank you for submitting your manuscript to PLOS ONE. After careful consideration, we feel that it has merit but does not fully meet PLOS ONE’s publication criteria as it currently stands. Therefore, we invite you to submit a revised version of the manuscript that addresses the points raised during the review process.

We look forward to receiving your revised manuscript.

Kind regards,

Neetu Choudhary, PhD

Academic Editor

PLOS ONE

Journal Requirements:

Reviewers' comments:

Reviewer's Responses to Questions

**Comments to the Author**

1. Does the manuscript provide a valid rationale for the proposed study, with clearly identified and justified research questions?

Reviewer #1: Yes

Reviewer #2: Yes

2. Is the protocol technically sound and planned in a manner that will lead to a meaningful outcome and allow testing the stated hypotheses?

Reviewer #1: Yes

Reviewer #2: Yes

3. Is the methodology feasible and described in sufficient detail to allow the work to be replicable?

Reviewer #1: Yes

Reviewer #2: Yes

4. Have the authors described where all data underlying the findings will be made available when the study is complete?

Reviewer #1: Yes

Reviewer #2: No

5. Is the manuscript presented in an intelligible fashion and written in standard English?

Reviewer #1: Yes

Reviewer #2: Yes

6. Review Comments to the Author

You may also provide optional suggestions and comments to authors that they might find helpful in planning their study.

Reviewer #1: This is an interesting study on “Methodology for developing and validating Bangladesh Healthy Eating Index: A Study protocol”.

My comments are appended.

Abstract:

Abstract has been written in right direction.

Introduction:

Introduction has been written in right direction.

Methods:

Line 93: Weighing, standard, and scoring: The author needs to put some arguments, why all eight groups of foods are given equal weight?

Line 138: How the sample size was calculated needs to be mentioned. Is this sample size 1080 representative to whole Bangladesh? Ensuring proper sample size, the author need to ensure external validity of the study.

Line 143: The mentioned correlation between dietary pattern and food security with cropping pattern and agricultural productivity may not be true in all areas like in urban settings. Moreover, due to socioeconomic, demographic and nutrition transition, availability of processed foods has become easily accessibly in rural areas, while rural area is higher productive in terms of agriculture. Moreover, people from agriculturally productive areas may sell their produce, rather than consuming, to afford other household commodities. For instance, farmer producing vegetables does not necessarily ensure that they will consume more vegetables. The author may reconsider their sampling procedure to make their study more representative to whole Bangladesh.

Line 176: The author mentioned that two consecutive days 24h dietary data of will be collected. 24H data provide short term dietary intake pattern. For long term dietary intake pattern, it is better to include FFQ and to compare 24H data with FFQ.

Reviewer #2: Summary

In this study, authors describe the methodological approach to the development and validation of a healthy eating index specific for Bangladeshi population. The index is based on the food-based dietary guidelines of Bangladesh and includes 8 components. For evaluation, diet will be assessed using repeated 24-h dietary recalls among 1080 reproductive age women. Developing an index tailored to dietary recommendations is relevant and the analysis plan for evaluation is well described. However, the usual intake estimation approach may not be adequate for the intended analysis (see first major comment). Also, I think it would be relevant to better describe how the score may be revised or adapted given the evaluation results (see second major comment).

Major

Usual intake estimation (L185). To my knowledge, the MSM procedure does not support bivariate analysis (e.g., joint measurement error correction of a score for a component and nutrient intake) or multivariate analysis (joint measurement error correction for the 8 components of the score to calculate the total score and nutrient intake). If it is the case, this is a major limitation of the planned usual intake analysis. Bias is expected. A state-of-the-art analysis would also use methods such as the National Cancer Institute multivariate Markov Chain Monte Carlo (MCMC) algorithm (Zhang, Krebs-Smith, et al., 2011; Zhang, Midthune, et al., 2011). The MCMC method would allow authors to generate measurement-error corrected data suitable to estimate usual intake distribution, multidimensionality and internal consistency analysis. This would not be possible with the MSM method.

In table 2, authors state their hypothesis (“Expectation”) regarding the evaluation. I think it would be useful to expand on 1) the rationale for these hypotheses; 2) the possibility that the hypotheses are *not* verified. For example, what if the score does not show many variations for a given component? How would authors approach or revise their scoring? What if the score is highly dependent on energy intake (see specific minor comments below)? What if some components of the score are inversely associated and show poor internal consistency? All in all, pre-specifying how the components or the scoring may be modified given the evaluation results will allow authors to have more flexibility once the evaluation results are known. Otherwise, there is a risk that authors find poor evaluation metrics and are stuck with a score that is not useful.

Minor

Scoring for the optimum component. This scoring approach sometimes results in poor internal consistency and may not “work” well within a score. Authors should examine or consider alternative scoring approach if the optimum components show poor results during the evaluation analysis. To be clear, I am not saying that authors should not use the “optimum scoring approach”, but rather consider that the approach may not work well within a total score. Expanding the discussion or method section regarding this possibility may be relevant.

Data analysis plan. The sampling strategy may require the use of survey-specific procedures or statistical analyses instead of standard procedures. I recommend authors mention how the sampling design will be considered, e.g., stratification, clustering, weighting accounted for using bootstrap or BRR variance estimation and sampling weights.

For construct validity, it would be relevant to add the (expected) relationship between sociodemographic characteristics and scores. For example, older adults typically have greater diet quality than younger adults and non-smoker typically have greater diet quality than smokers, at least in a North American context (Reedy et al., 2018). Authors could also select other characteristics that are more relevant to their context.

Authors expect a weak correlation between energy intake and the score (L216, table 2). However, many components are scored in the absolute scale (e.g., fruits >= 1 serving/day or sugar <= 5 servings/day). Typically, scoring on the absolute scale *is* associated with energy intake, because the amount of food eaten influence the probability of eating more or less than (absolute) thresholds. While the evaluation may show that this is not a major issue for the score per se, I think authors should further describe why they expect a weak correlation or revise their hypothesis. Also, the use of an absolute approach vs. a relative scoring approach (e.g., % of total energy as in the US-HEI) should be discussed.

References

Reedy, J., Lerman, J. L., Krebs-Smith, S. M., Kirkpatrick, S. I., Pannucci, T. E., Wilson, M. M., Subar, A. F., Kahle, L. L., & Tooze, J. A. (2018). Evaluation of the Healthy Eating Index-2015. J Acad Nutr Diet, 118(9), 1622-1633. https://doi.org/10.1016/j.jand.2018.05.019

Zhang, S., Krebs-Smith, S. M., Midthune, D., Perez, A., Buckman, D. W., Kipnis, V., Freedman, L. S., Dodd, K. W., & Carroll, R. J. (2011). Fitting a bivariate measurement error model for episodically consumed dietary components. Int J Biostat, 7(1), 1. https://doi.org/10.2202/1557-4679.1267

Zhang, S., Midthune, D., Guenther, P. M., Krebs-Smith, S. M., Kipnis, V., Dodd, K. W., Buckman, D. W., Tooze, J. A., Freedman, L., & Carroll, R. J. (2011). A New Multivariate Measurement Error Model with Zero-Inflated Dietary Data, and Its Application to Dietary Assessment. Ann Appl Stat, 5(2B), 1456-1487. https://doi.org/10.1214/10-AOAS446

7. PLOS authors have the option to publish the peer review history of their article (what does this mean?). If published, this will include your full peer review and any attached files.

Reviewer #1: **Yes: **Md Kamruzzaman

Reviewer #2: No

---

## [Author Response · Author response to Decision Letter 0]

26 Jul 2024

Reviewer 1:

Comment 1: Line 93: Weighing, standard, and scoring: The author needs to put some arguments, why all eight groups of foods are given equal weight.

Response: Since all the components are equally important in the dietary guideline, each component will receive equal weightage. A similar approach was followed while updating US HEI-2015 (Reference 11): Page 5-6.

Comment 2: Line 138: How the sample size was calculated needs to be mentioned. Is this sample size 1080 representative to whole Bangladesh? Ensuring proper sample size, the author need to ensure external validity of the study.

Line 143: The mentioned correlation between dietary pattern and food security with cropping pattern and agricultural productivity may not be true in all areas like in urban settings. Moreover, due to socioeconomic, demographic and nutrition transition, availability of processed foods has become easily accessibly in rural areas, while rural area is higher productive in terms of agriculture. Moreover, people from agriculturally productive areas may sell their produce, rather than consuming, to afford other household commodities. For instance, farmer producing vegetables does not necessarily ensure that they will consume more vegetables. The author may reconsider their sampling procedure to make their study more representative to whole Bangladesh.

Response: We are planning to follow cluster sampling and sample size calculation procedure is elaborated in Figure 3 and Page 10. 

It has already been mentioned under the “limitation” section that the study wouldn’t cover the whole of Bangladesh and therefore further study is needed. It would be best if nationally representative dietary survey data could be utilized. But due to a lack of resources, we can’t plan to conduct a nationally representative survey at this moment. Instead, we are planning to establish a methodological approach for developing a healthy eating index and validity will be checked at a smaller scale. If it works, we have a plan to apply the procedure to a large-scale dietary survey. 

However, in this small-scale survey, we also consider rural-urban issues and agricultural productivity. The sample will be taken from both rural and urban areas. In addition, both high and low-productive areas will be selected.

Comment 3: The author mentioned that two consecutive days 24h dietary data of will be collected. 24H data provide short term dietary intake pattern. For long term dietary intake pattern, it is better to include FFQ and to compare 24H data with FFQ.

Response: It is not the scope of the study to compare the two dietary data collection approaches: 24H recall Vs FFQ. There are ample of evidences on this issue. 

Research has indicated that short-term instruments, such as 24-hour dietary recalls, tend to provide less-biased estimates of dietary intake than tools that query usual intake directly, such as food frequency questionnaires (FFQs) (National Cancer Institute, 2015). Therefore, use of a short-term measure is the preferred method for estimating usual, or long-run average, dietary intake.

When estimating distributions of usual intake of foods, food groups, or nutrients, including FFQ information does not appear to have a large impact on estimated values (Tooze et al., 2010; Goedhart et al., 2012). Therefore, it is expected that use of 24-hour recalls alone should be sufficient to estimate usual intakes for estimating distributions of food, food group, and nutrient intake to assess a population’s intake or inform food fortification policies. Most commonly, the FFQ data are included as a covariate in a statistical model and, therefore, while beneficial to improve precision, are not required to obtain an unbiased estimate of the diet-health relationship.

In addition, 24-hour recall is a valid and reliable dietary data collection tools for Low and middle income countries. There is also no validated FFQ for Bangladesh. Hence, we are planning to apply 24 hour recall method.

Reviewer 2: 

Major 1: Usual intake estimation (L185). To my knowledge, the MSM procedure does not support bivariate analysis (e.g., joint measurement error correction of a score for a component and nutrient intake) or multivariate analysis (joint measurement error correction for the 8 components of the score to calculate the total score and nutrient intake). If it is the case, this is a major limitation of the planned usual intake analysis. Bias is expected. A state-of-the-art analysis would also use methods such as the National Cancer Institute multivariate Markov Chain Monte Carlo (MCMC) algorithm (Zhang, Krebs-Smith, et al., 2011; Zhang, Midthune, et al., 2011). The MCMC method would allow authors to generate measurement-error corrected data suitable to estimate usual intake distribution, multidimensionality and internal consistency analysis. This would not be possible with the MSM method

Response: Thanks a lot for your valuable suggestion. We have updated the statements accordingly and mention that MCMC method will be applied (Page 12). 

Major 2: In table 2, authors state their hypothesis (“Expectation”) regarding the evaluation. I think it would be useful to expand on 1) the rationale for these hypotheses; 2) the possibility that the hypotheses are *not* verified. For example, what if the score does not show many variations for a given component? How would authors approach or revise their scoring? What if the score is highly dependent on energy intake (see specific minor comments below)? What if some components of the score are inversely associated and show poor internal consistency? All in all, pre-specifying how the components or the scoring may be modified given the evaluation results will allow authors to have more flexibility once the evaluation results are known. Otherwise, there is a risk that authors find poor evaluation metrics and are stuck with a score that is not useful. 

Response: In table 2, the rationale for the “expected outcome” is added. Alongside, we have elaborated the discussion in the “method” and “discussion” section including alternative approaches (See Table 2, Page 14-15, 23)

Minor 1: Scoring for the optimum component. This scoring approach sometimes results in poor internal consistency and may not “work” well within a score. Authors should examine or consider alternative scoring approach if the optimum components show poor results during the evaluation analysis. To be clear, I am not saying that authors should not use the “optimum scoring approach”, but rather consider that the approach may not work well within a total score. Expanding the discussion or method section regarding this possibility may be relevant. 

Response: We have discussed about the alternative approaches under “Discussion” section (Page 22-23)

Minor 2: Data analysis plan. The sampling strategy may require the use of survey-specific procedures or statistical analyses instead of standard procedures. I recommend authors mention how the sampling design will be considered, e.g., stratification, clustering, weighting accounted for using bootstrap or BRR variance estimation and sampling weights. 

Response: A stratified two-step cluster sampling approach will be used for the sampling design, with district agricultural productivity serving as the basis for stratification and district, area, and EA-level clustering. It is anticipated that each stratum and cluster will have equal odds of selection in this sampling design. As a result, straightforward random sampling inside each stratum or cluster is presumed, and intricate weighting modifications are not required. Each sampled household will receive sampling weights since a two-step cluster sampling strategy will be used. Considering the stratification and clustering in the design, the sampling weight for every household will be computed as the inverse of its likelihood of selection. The survey results can then be appropriately represented to the target population by using these sampling weights in the analysis.

Minor 3: For construct validity, it would be relevant to add the (expected) relationship between sociodemographic characteristics and scores. For example, older adults typically have greater diet quality than younger adults and non-smokers typically have greater diet quality than smokers, at least in a North American context (Reedy et al., 2018). Authors could also select other characteristics that are more relevant to their context. 

Response: Variation of HEI across various socio-demographic groups: Analyses will focus on whether the participants’ socio-demographic characteristics differ significantly throughout the tertiles of the Bangladesh Healthy Eating Index (HEI), including changes in district distribution, education levels, household food security status, dietary diversity, and wealth index. We will also look at the mean HEI score (both total score and individual component score) among different socio-demographic group. We also expect higher HEI score among women with higher dietary diversity, household food security, higher level of education, and higher wealth index. Reason for such expectation that previous studies reported better diet quality among these socio-economic groups. (see Table 2 and page 14).

Minor 4: Authors expect a weak correlation between energy intake and the score (L216, table 2). However, many components are scored in the absolute scale (e.g., fruits >= 1 serving/day or sugar <= 5 servings/day). Typically, scoring on the absolute scale *is* associated with energy intake, because the amount of food eaten influence the probability of eating more or less than (absolute) thresholds. While the evaluation may show that this is not a major issue for the score per se, I think authors should further describe why they expect a weak correlation or revise their hypothesis. Also, the use of an absolute approach vs. a relative scoring approach (e.g., % of total energy as in the US-HEI) should be discussed.

Response: We are expecting a weak association between HEI score and energy intake. However, many components are scored in the absolute scale (such as fruits, sugar) and scoring on the absolute scale might show strong association with energy intake, because the amount of food eaten influence the probability of eating more or less than (absolute) thresholds. By using a density method, the relationship between energy intake and diet quality indices is weakened [11,26], resulting in an indicator that is less dependent on the quantity of food consumed. A higher sensitivity to detect changes over time because of fewer “ceiling” and “floor” effects and uniformity of scoring standards among index components—which can be computed as ratios of all foods or beverages consumed or as a percentage of total energy (%E)—are further benefits of employing the density approach. Therefore, these dietary components could also be evaluated using relative methods in addition to absolute ones, such as percentage of total energy or food group intake per 1000 kcal (see page 23-24).

---

## [Decision Letter · Decision Letter 1]

7 Aug 2024

Methodology for developing and validating Bangladesh Healthy Eating Index: A Study protocol

PONE-D-24-15594R1

Dear Dr. Islam,

We’re pleased to inform you that your manuscript has been judged scientifically suitable for publication and will be formally accepted for publication once it meets all outstanding technical requirements.

Kind regards,

Muttaquina Hossain, MBBS, MPH

Academic Editor

PLOS ONE

Additional Editor Comments (optional):

Reviewers' comments:

Reviewer's Responses to Questions

**Comments to the Author**

1. Does the manuscript provide a valid rationale for the proposed study, with clearly identified and justified research questions?

Reviewer #1: Yes

Reviewer #2: Yes

2. Is the protocol technically sound and planned in a manner that will lead to a meaningful outcome and allow testing the stated hypotheses?

Reviewer #1: Yes

Reviewer #2: Yes

3. Is the methodology feasible and described in sufficient detail to allow the work to be replicable?

Reviewer #1: Yes

Reviewer #2: Yes

4. Have the authors described where all data underlying the findings will be made available when the study is complete?

Reviewer #1: Yes

Reviewer #2: Yes

5. Is the manuscript presented in an intelligible fashion and written in standard English?

Reviewer #1: Yes

Reviewer #2: Yes

6. Review Comments to the Author

You may also provide optional suggestions and comments to authors that they might find helpful in planning their study.

Reviewer #1: Thanks for your feedback and properly address issues raised by the reviewers. I hope the manuscript is now suitable for publication.

Reviewer #2: I thank the authors for diligently addressing my previous comments. Thank you for this work. I have no additional feedback to provide.

Please note that in addition to the currently available SAS version of the NCI MCMC algorithm ( https://prevention.cancer.gov/research-groups/biometry/measurement-error-impact/software-measurement-error/several-regularly-consumed-or-episodically-consumed-foods-or-nutrients-multivariate-distribution ), an R version of the MCMC algorithm should be made available by the NCI before completion of the study. Authors may be interested in this version, since it will be based on the open-source R software.

7. PLOS authors have the option to publish the peer review history of their article (what does this mean?). If published, this will include your full peer review and any attached files.

Reviewer #1: **Yes: **Md Kamruzzaman

Reviewer #2: No

---

## [Editor Report · Acceptance letter]

13 Aug 2024

PONE-D-24-15594R1 

PLOS ONE

Dear Dr. Islam, 

I'm pleased to inform you that your manuscript has been deemed suitable for publication in PLOS ONE. Congratulations! Your manuscript is now being handed over to our production team.

Kind regards, 

on behalf of

Dr. Muttaquina Hossain 

Academic Editor

PLOS ONE